# Research Progress on the Degradation of Human Milk Oligosaccharides (HMOs) by Bifidobacteria

**DOI:** 10.3390/nu17030519

**Published:** 2025-01-30

**Authors:** Ruitao Cai, Jie Zhang, Yingte Song, Xiaoyong Liu, Huilian Xu

**Affiliations:** School of Biological Science and Technology, University of Jinan, Jinan 250024, China; crt17853319565@163.com (R.C.); jie_zhang1229@163.com (J.Z.); 202321201498@stu.ujn.edu.cn (Y.S.)

**Keywords:** Bifidobacteria, HMOs, degradation mechanism, infant nutrition

## Abstract

The purpose of this study was to investigate the degradation mechanism of Bifidobacterium on breast milk oligosaccharides (HMOs) and its application in infant nutrition. The composition and characteristics of HMOs were introduced, and the degradation mechanism of HMOs by Bifidobacterium was described, including intracellular and extracellular digestion and species-specific differences. The interaction between Bifidobacterium and Bacteroides in the process of degrading HMOs and its effect on intestinal microecology were analyzed. The effects of HMO formula milk powder on the intestinal microbiota of infants were discussed, including simulating breast milk composition, regulating intestinal flora and immune function, infection prevention, and brain development. Finally, the research results are summarized, and future research directions are proposed to provide directions for research in the field of infant nutrition.

## 1. Introduction

HMOs are a variety of complex carbohydrates that play a key role in the growth and development of infants. They are the third most abundant solid component in breast milk, with more than 200 different types, providing a variety of health benefits, such as promoting digestive health, regulating immune response, and promoting cognitive development [1]. Bifidobacterium is one of the most abundant microbial genera in the intestine of breastfed infants and plays an important role in the metabolism of HMOs. Breast milk is the gold standard of infant nutrition, which is rich in HMOs [2]. The purpose of this study was to explore the mechanism of HMO degradation by Bifidobacterium and its application in infant nutrition.

An HMO is a general term for an oligosaccharide with a degree of polymerization no higher than 3 and that are naturally present in breast milk [3]. HMOs are composed of glucose, galactose, *N*-acetylglucosamine, fucose, and *N*-acetylneuraminic acid-modified lactose molecules [4]. At present, more than 200 different HMO components have been identified in breast milk, and their types and contents are affected by many factors such as genetic factors, geographical factors, gestational age, production methods, and lactation stages [5].

HMO plays an irreplaceable role in infant intestinal health. In the construction of intestinal flora, after the birth of the baby, the intestine is a relatively sterile environment, which needs to quickly establish a beneficial microbial community. HMO can accurately provide nutrients for the growth of beneficial bacteria, such as bifidobacteria, and stimulate their mass reproduction [6]. For example, bifidobacteria longum and bifidobacteria shortum can use HMOs for growth and metabolism, thus, forming dominant flora in the intestine. These beneficial bacteria inhibit the growth and colonization of harmful bacteria, such as Escherichia coli and Salmonella, by competing for nutrients and producing antibacterial substances, maintaining the balance of intestinal microecology, and effectively reducing the risk of intestinal infection in infants [7].

Bifidobacterium is one of the most active species in the metabolism of HMOs in the intestinal flora of infants. Different bifidobacteria can selectively use different types of HMOs as metabolic substrates, suggesting that the combined application of different HMOs may promote the colonization of specific bifidobacteria, thereby helping to build a healthy intestinal microecology [6]. For example, Bifidobacterium bifidum and Bifidobacterium longum subsp. infantis are the two strains that use the most types of HMOs, and they can use HMOs with different structures modified by different groups [7]. The utilization of HMOs by Bifidobacterium has species-specific differences. Infant intestinal-associated Bifidobacterium has evolved two ways to use HMOs [8].

By degrading HMO, Bifidobacterium can convert it into metabolites such as short-chain fatty acids (such as lactic acid and acetic acid). These metabolites not only provide energy for intestinal epithelial cells but also reduce intestinal pH and inhibit the growth of harmful bacteria, thereby maintaining the balance of intestinal flora. At the same time, it is of great significance to the development of the infant immune system. Studies have shown that bifidobacteria produce metabolites, such as aromatic lactic acid, during the degradation of HMO. These substances can regulate the function of immune cells, enhance the immune response of infants, and reduce the risk of infection [9]. Bifidobacterium can also increase the production of secretory immunoglobulin A (sIgA) by promoting the colonization of intestinal beneficial bacteria and further enhancing the immune barrier function of intestinal mucosa [10]. In the process of breastfeeding, the efficient use of HMO by Bifidobacterium makes it occupy a competitive advantage in the infant’s intestine, which helps infants better adapt to breast milk nutrition [11]. This symbiotic relationship not only optimizes the intestinal environment but also plays a key role in the early intestinal barrier function and immune system development of infants [12].

In addition, the ability of different Bifidobacterium species to utilize HMOs varies significantly. Bifidobacterium longum subsp. infantis (B. infantis) is renowned for its exceptional capacity to degrade HMOs, making it one of the most proficient HMO utilizers among gut bacteria. This subspecies possesses a diverse array of glycoside hydrolase enzymes that enable it to break down a wide range of HMO structures, including fucosylated and sialylated forms. While other Bifidobacterium species, such as B. bifidum, also exhibit the ability to utilize HMOs, their capacity is generally lower compared to B. infantis. However, they can still degrade HMOs into monosaccharides, contributing to the overall metabolic processes in the infant gut [13]. Other genera of the neonatal gut microbiota, such as Clostridium, Enterococcus, Escherichia coli, Staphylococcus, and Streptococcus, do not degrade HMOs themselves, but they may use some of the decomposition products or fermentation end products produced by other intestinal bacteria [14]. Colon microbiota fermentation of HMOs can produce beneficial metabolites, such as short-chain fatty acids (SCFAs) [15]. The intestinal bacteria that can produce SCFAs are Bacteroides, Bifidobacterium, Lactobacillus, etc. SCFAs play a key role in the communication between the intestinal bacterial community and the host, and they are essential for the intestinal health of newborns [16]. The main product of fermentation is acetic acid, which reduces the pH value in the intestine, and it has a bacteriostatic effect and can inhibit the growth of pathogenic bacteria. In addition to acetic acid, the fermentation products also include butyric acid and propionic acid, which can interact with host epithelial cells, stimulate mucin release, increase mucosal blood flow, and regulate the immune system [17]. Lactic acid and succinic acid are intermediate metabolites in the production of SCFAs, but there are few studies [18].

## 2. The Composition and Properties of HMOs

### 2.1. Monosaccharide Composition of HMOs

HMOs are naturally occurring oligosaccharides with a degree of polymerization not higher than 3 in breast milk, and their monosaccharide composition is rich and diverse [16]. According to the different structural types, HMOs are mainly divided into neutral fucosylated HMOs, such as 2′-fucosyllactose (2′-FL) and 3-fucosyllactose (3-FL), neutral non-fucosylated HMOs, such as lacto-N-tetraose (LNT), and lacto-N-neotetraose (LNnT), and acidic sialylated HMOs. Such as 3′-sialyllactose (3′-SL) and 6′-sialyllactose (6′-SL). After ingestion of HMOs, about 0.05% of HMOs enter the blood, and most of the remaining HMOs enter the large intestine. Observational studies have shown that HMOs play an important role in improving intestinal flora microecology, maintaining intestinal barrier, regulating immunity, resisting pathogen infection, and promoting neural development. It is mainly modified by five monomers: glucose, galactose, *N*-acetylglucosamine, fucose, and *N*-acetylneuraminic acid. Among them, D-glucose (Glc) is a common monosaccharide that plays an important structural support role in HMOs. D-galactose (Gal) is similar to glucose in structure and participates in the construction of complex structures of HMOs. *N*-acetylglucosamine (GlcNAc) brings unique chemical properties to HMOs [17]. L-fucose (Fuc) endows HMOs with specific biological functions, while *N*-acetylneuraminic acid (Neu5Ac) in sialic acid (Sia) is the most important form of sialic acid and plays a key role in the function of HMOs [18]. These monosaccharides are linked to lactose through different glycosidic bonds to form different structures of HMOs. Each HMO contains a lactose at the reducing end, and on this basis, the galactose β-1,3-N-acetylglucosamine is connected by β-1,3 or β-1,6 bonds, or the *N*-acetylgalactosamine is connected to extend the sugar chain to form core structures such as lacto-N-tetrose (LNT) and lacto-N-neotetraose (LNnT) [19]. Breast milk oligosaccharides are complex mixtures of many oligosaccharides, and more than 200 HMO structures have been isolated and identified [20].

The structure of some HMOs is shown in Figure 1.

### 2.2. Performance of HMOs

The composition of HMOs in breast milk is complex and diverse, and the structure and content of its main components are affected by many factors. The concentration of HMOs in the milk of secretory mothers was higher than that of non-secretory mothers [21]. Specifically, the total HMO concentration of secretory mothers was about 15.91 ± 2.80 μmol/mL, while the total HMO concentration of non-secretory mothers was about 8.94 ± 1.51 μmol/mL [4]. About 79% of mothers are secretory mothers. The structure of HMOs is based on lactose molecules, and galactose β-1,3-N-acetylglucosamine (Gal-β-1,3-GlcNAc) or β-1,6-glycosidic linkage *N*-acetyllactose (Gal-β-1,4-GlcNAc) is extended outward through β-1,3-glycosidic linkage on the lactose core (Gal-β-1,4-Glc) [22]. At the same time, these core structures can be further modified with fucose or sialic acid to form slender linear structures or branched oligosaccharides. The main components of HMOs include neutral fucosylated HMOs, neutral non-fucosylated HMOs, and acidic sialylated HMOs [23]. Among them are neutral fucosylated HMOs such as 2′-fucosylated lactose (2′-FL) and 3-fucosylated lactose (3-FL); neutral non-fucosylated HMOs such as lactose-N-tetrasaccharide (LNT) and lactose-N-neotetrasaccharide (LNnT); and acidic sialylated HMOs such as 3′-sialyllactose (3′-SL) and 6′-sialyllactose (6′-SL) [24].

The content of different types of HMOs in breast milk is different [25], and its content is also affected by the mother’s secretion type, gene, body mass index, gestational age, dietary habits, geographical environment, and other factors. For example, secretory genes (Se or FUT2) and Lewis genes (Le or FUT3) express α-1,2-fucosyltransferase and α-1,3/4-fucosyltransferase, respectively, resulting in higher concentrations of α-1,2-fucosylated HMOs such as 2′-fucosyllactose (2′-FL) and lacto-N-fucopentaose (LNFP) I in Se/Le and Se/Le-secretory breast milk, rather than the lack of such HMOs in non-secretory breast milk [26].

The reviewed scientific literature shows that the main components of HMOs play an important role in child growth and development. The structure and content of the main components of HMOs are complex and diverse. Understanding the composition and content of HMOs is helpful to further study its application in infant nutrition and health.

## 3. The Degradation Mechanism of HMOs by Bifidobacterium

Bifidobacteria play an important role in the metabolism of HMOs in infant gut microbiota. It has been found that these bacteria have a variety of genes that encode glycosidases and transporters related to HMOs metabolism [27]. *B. longum subsp* strains possess a 43 kb gene cluster specific for HMO degradation [28]. This gene cluster encodes all GH enzymes required to efficiently cleave HMOs, including 1,2-α-fucosidases belonging to the GH95 family, 1,3/4-α-fucosidases from GH29, 2,3/6 sialidases belonging to the GH33 family, β-N-acetylhexosaminidase enzymes from the GH20 family, β-galactosidases from the GH2 family, and LNT β-galactosidases from the GH42 family [28,29]. Interestingly, these enzymes appear to be intracellular due to the absence of an *N*-terminal signal sequence in their primary structure [28]. This gene cluster also includes several genes that encode sugar transporters involved in importing HMOs intact such as ATP-binding cassette (ABC) transporters, GNB/LNB pathway transporters, as well as solute binding proteins (SBPs) [30]. ABC transporters recognize and bind to HMOs through their extracellular solute-binding proteins (SBPs). These SBPs have high substrate specificity and can bind to the specific structure of HMOs and transmit them to the membrane protein complex. When SBPs bind to HMOs, the membrane protein portion of ABC transporters undergoes conformational changes, binds to ATP, and hydrolyzes ATP to release energy, thereby transporting HMOs into cells. This transport process not only depends on the high specificity of ABC transporters but is also closely related to the structural complexity of HMOs [31]. SBPs interact with specific glycan structures in HMOs through their binding sites, thereby realizing the recognition and binding of HMOs. This binding depends on the highly specific structure of SBPs, enabling them to form stable interactions with specific glycans such as fucose, sialic acid, or lactose in HMOs. During the binding process, the structure of SBPs will undergo conformational changes, which will help to further enhance the binding stability with HMOs. This high affinity binding allows SBPs to introduce intact HMO molecules into cells through ABC transporters [32]. As shown in Figure 2, Bifidobacterium has developed two mechanisms for the use of HMOs: one is digestion in cells through transporters; the other is dependent on extracellular glycosidase digestion in the extracellular [28]. For example, B.bifidum and some B.longum subsp. longum strains lack complete HMO transporters, so they mainly rely on extracellular glycosidases to decompose HMOs outside the cell to obtain monosaccharides or disaccharides, while other HMO degradation products enter the cell through transporters for further decomposition [29]. In 2008, the genome sequence of B.longum subsp. infantis standard strain ATCC 15697 was published, revealing a number of genes related to the potential adaptability of B.longum subsp. infantis to infant hosts, including gene clusters related to HMOs catabolism, extracellular solute-binding proteins, and permeases that are predicted to be active against HMOs [30].

In order to degrade HMOs well, Bifidobacterium has evolved a variety of glycosidases and phosphorylases with high specificity for HMOs [31]. Through the analysis of the whole genome sequence of B.longum subsp. infantis, the researchers excavated a variety of glycosidase genes involved in the utilization of HMOs. These gene families are highly conserved and cover the function of hydrolyzing almost all the linkages in the structure of HMOs [32]. Fucosidases are responsible for the removal of fucose from HMOs, including 1,2-α-L-fucosidase and 1,3/4-α-L-fucosidase, which have different sensitivities to HMOs with different structures. The 1,2-α-L-fucosidase is highly sensitive to Fucα1-2Gal-O-R and can recognize 2′-FL and LNFP I, and it also has certain activity to LDFT, LNDFH I, and 3-FL. The 1,3/4-α-L-fucosidase requires a branched galactose residue to hydrolyze the fucosidic bond, acting on LNFP II and LNFP III [33].

The neuraminidase NanH2 is responsible for the separation of NeuAc from the core structure and has a good effect on the hydrolysis of α-2,3 and α-2,6 glycosidic bonds. After removing the modifiers, β-galactosidase can hydrolyze the core structure of HMOs, including LNTβ-1,3-galactosidase and β-1,4-galactosidase. This enzyme separates NeuAc from the core structure of HMOs and acts on α-2,3 and α-2,6 glycosidic bonds in sialylated HMOs [34].

The β-N-acetylglucosaminidase has a strong ability to hydrolyze LNTri II and is also active for GlcNAc residues linked to β-1,6 bonds in LNH. GNB/LNB phosphorylase is an intracellular enzyme that can reversibly phosphorylate GNB/LNB to produce α-galactose 1-phosphate (Gal1P) and *N*-acetylgalactosamine (GalNAc)/GlcNAc, saving ATP consumption. After the removal of modifications by fucosidase and NanH2, β-galactosidase hydrolyzes the core structure of HMOs. LNTβ-1,3-galactosidase specifically hydrolyzes LNT into Gal and LNTri II, with the strongest activity against LNT, followed by Lac, LNB, and LNnT. The β-1,4-galactosidase acts on Lac and type II chains (Galβ1-4GlcNAc-O-R) [35].

In addition to sugar degradation genes, bifidobacteria also need related transporters to ensure their growth ability in the presence of HMOs. B.longum subsp. infantis can introduce intact HMOs into cells through the mediation of several SBPs. B.longum subsp. infantis ATCC 15697 is a typical strain with a strong ability to utilize HMOs [36]. This strain has a gene cluster that encodes several intracellular HMO-related glycosidases and SBPs for adenosine triphosphate binding cassette transporters (ABC) [37]. Table 1 lists the different enzymes involved in the degradation of HMOs. The GNB/LNB transporter is responsible for the transport of LNB released extracellularly by lactate-N-bioenzymes and GNB released from mucin O-glycans by α-N-acetylgalactosaminidase. FL transporters can transport 2′-FL, 3-FL, LDFT, and LNFP I into cells [38]. B.longum subsp. infantis ATCC 15697 has two homologous FL transporters, and its SBPs have 60% consistency. The LNnT transporter is responsible for the transport of LNnT by NahS (LNnT-BP). Although glycosidases are well understood, there are relatively few studies on transporters. At present, only the transporters of LNB, FL, and LNnT have been characterized [39].

## 4. Effects of HMOs on Gut Microbiota

### 4.1. The Effect of HMO on Intestinal Tract

As the energy source of intestinal beneficial bacteria, HMOs provide a key material basis for the growth and metabolism of beneficial bacteria such as bifidobacteria and lactic acid bacteria [43]. A series of specific glycoside hydrolases and transporters are distributed on the surface of Bifidobacterium. Like precise ‘molecular tools’, they can specifically recognize and efficiently bind HMO molecules [44]. Through the catalysis of glycoside hydrolases, the complex sugar chain structure of HMOs is gradually dismantled and converted into monosaccharides such as glucose and galactose, as well as easily absorbed metabolites such as short-chain fatty acids (SCFAs) [28]. Studies have found that Bifidobacterium infantis in newborns can inhibit the growth of harmful microorganisms in the intestine and help infants digest breast milk oligosaccharides (HMO) in breast milk. When HMO enters the infant intestine, Bifidobacterium infantis can convert it into short-chain fatty acids, thereby effectively reducing the pH value of infant feces and creating an acidic environment that is not conducive to the growth of harmful bacteria. As the baby grows, this probiotic will gradually decrease, but if it does not appear in the early stage, other bacteria, especially pathogens, may fill its growth space, further increasing the probability of infants suffering from related diseases [45]. Researchers analyzed breast milk oligosaccharides in 33 children with necrotizing enterocolitis (NEC) and 37 controls and performed longitudinal metagenomic sequencing of feces from 48 infants, including 14 children with NEC. The results showed that the concentrations of HMO and DSLNT in breast milk of mothers with NEC were significantly lower than those in the control group. The sensitivity and specificity of MOM threshold of 241 nmol/mL for NEC was 0.9. The metagenomic sequencing before the onset of NEC showed that the relative abundance of Bifidobacterium longum decreased and the relative abundance of Enterobacter cloacae increased in children with NEC. Low MOM DSLNT affects the longitudinal development of the microbiome, which is associated with a decrease in the type of gut microbiota in preterm infants, transitioning to a bifidobacteria-dominated microbiome in preterm infants, usually observed in older infants. By combining pre-disease HMO and metagenomic data with random forest analysis, the accuracy of distinguishing infant health or NEC reached 87.5% [46]. The researchers compared the composition of intestinal flora between cesarean section and natural delivery infants and found that the abundance of beneficial bacteria in the intestinal flora of cesarean section infants was lower, while the relative proportion of harmful bacteria was higher. Further studies have found that in the breastfed cesarean section infants, if breast milk contains abundant HMOs such as 2′-FL, the structure and function of the infant’s intestinal flora will be closer to that of natural birth infants. 2′-FL can promote the growth and colonization of beneficial bacteria such as Bifidobacterium. Bifidobacteria produce short-chain fatty acids by fermenting HMO, reduce intestinal pH, inhibit the growth of harmful bacteria, and regulate the balance of intestinal flora [47]. In the study of children with milk protein allergy (CMPA), deep hydrolyzed formula milk powder supplemented with HMO was used for feeding. Studies have found that HMO can regulate intestinal flora, promote the enrichment of beneficial bacteria such as Bifidobacterium, and reduce the number of potentially harmful bacteria such as Escherichia coli. For children with CMPA who began to be fed with a deeply hydrolyzed formula (eHF) containing HMO before 3 months of age, this regulatory effect is more obvious, which helps to reverse the state of intestinal flora imbalance [48]. In related studies, the intestinal environment was simulated by in vitro experiments, and common intestinal pathogens, such as Escherichia coli and Salmonella, were co-cultured with culture medium containing HMO. The results showed that HMOs could bind to the adhesion protein on the surface of pathogens and block the adhesion of pathogens to intestinal epithelial cells, thus, effectively inhibiting the colonization and reproduction of pathogens in the intestine [49]. Researchers, by studying the isolated colonic smooth muscle tissue, applied different concentrations of fucose-based HMOs and used a tension sensor to accurately measure the changes in the contractility of the smooth muscle. The results showed that with the increase in fucosylated HMO concentration, the contractility of colonic smooth muscle gradually decreased, and this effect showed a certain dose-dependent manner [50].

HMOs have a strong inhibitory effect on harmful bacteria and can accurately combat the survival and reproduction of harmful bacteria through a variety of ways [47]. On the one hand, HMOs can pre-empt the binding of receptors on the surface of intestinal epithelial cells by virtue of their unique molecular structure and occupy the original adhesion sites of harmful bacteria, so that harmful bacteria such as Escherichia coli and Clostridium perfringens cannot be successfully attached to the intestinal mucosa, thus, blocking the first step of their invasion of intestinal tissue and causing inflammation [48]. It was found that in vitro cell experiments, intestinal epithelial cells were pre-incubated with a certain concentration of 2′-fucosyllactose (2′-FL), and then pathogenic E.coli was added. The number of E.coli adhesions was significantly reduced compared with the untreated group, with a decrease of more than 80%, effectively preventing the invasion of harmful bacteria [49].

On the other hand, HMOs are metabolized by beneficial bacteria in the intestine to produce short-chain fatty acids and other products, which can change the microenvironment in the intestine, such as reducing pH value, regulating redox potential, etc., creating a harsh living environment for harmful bacteria [50]. Taking butyric acid as an example, it can inhibit the gene expression of harmful bacteria such as Salmonella and Clostridium difficile, affect the synthesis and secretion of virulence factors, and weaken the pathogenicity of harmful bacteria [51]. At the same time, butyric acid can also promote the expression of tight junction proteins in intestinal epithelial cells, enhance intestinal barrier function, further block the invasion of harmful bacteria and their toxins, and protect intestinal health in an all-round way [2].

### 4.2. Regulating Intestinal Microbial Community Structure

HMOs can finely regulate and optimize the structure of intestinal microbial community [52]. In the critical period of intestinal development in infants and young children, HMOs in breast milk provide abundant nutrient substrates for beneficial bacteria such as bifidobacteria, promoting their rapid proliferation and becoming the dominant flora in the intestine [53]. A cohort study of Chinese infants showed that the relative abundance of bifidobacteria in the intestine of breastfed infants was more than 70% within 1 month after birth, while the abundance of bifidobacteria in formula-fed infants was relatively low, and the abundance of harmful bacteria such as Escherichia coli was relatively high [54]. Studies have shown that supplementation of HMOs with specific structures can increase the number of beneficial bacteria such as Akkermansia in the intestine. These beneficial bacteria can help alleviate intestinal inflammation and improve the symptoms of patients by regulating intestinal immune response and enhancing intestinal barrier function [55].

With the growth of beneficial bacteria such as Bifidobacterium, they exert selective pressure on the growth of other microorganisms in the intestine by secreting antibacterial substances and competing nutrients, guiding the intestinal microbial community to develop in a more healthy and stable direction and gradually building a balanced ecosystem dominated by beneficial bacteria (Table 2).

## 5. Effects of HMOs Formula Milk Powder on the Intestinal Tract and Brain of Infants

HMO formula milk powder has a positive effect on the intestinal flora of infants, which can promote the growth of beneficial bacteria, enhance the intestinal immune barrier, improve the diversity of intestinal flora, and may reduce the disease risk of infants [61]. As show in Table 3, the content of HMO in different formula milk powder. A multicenter, double-blind, randomized controlled trial included hundreds of healthy full-term infants, who were randomly divided into two groups and fed with infant formula with HMOs and ordinary formula without HMOs, respectively. The results showed that compared with the normal formula feeding group, the relative abundance of bifidobacteria in the intestinal tract of infants in the milk powder feeding group supplemented with HMOs increased significantly at 1 month after birth, with an average relative abundance of more than 60%, and it continued to maintain a high level in the following months. The relative abundance of potentially harmful bacteria such as Escherichia coli was significantly reduced by about 30–40% [62]. In terms of growth and development, there was no significant difference in weight and length growth between the two groups of infants, but the head circumference growth of infants in the HMOs group was slightly better than that in the control group, suggesting that it may promote brain development [63]. In terms of digestive function, the incidence of digestive discomfort symptoms such as vomiting and diarrhea in the HMOs group was reduced by about 20%, the baby’s defecation was more regular, and the fecal traits were closer to those of breastfed infants. In terms of infectious diseases, in the first 6 months after birth, the incidence of respiratory tract infection and gastrointestinal infection in the HMO group decreased by 1.2 times and 0.8 times, respectively, indicating that HMOs helped to enhance the immunity of infants and reduce the risk of infection [64]. In a study of IBD patients, after 12 weeks of oral administration of HMOs in the intervention group, the abundance of beneficial bacteria increased, harmful bacteria decreased, inflammatory indicators decreased, and clinical symptom scores decreased by 30%, indicating that HMOs can improve intestinal microecology and relieve symptoms [65]. In the study of obese patients, 100 obese adults (BMI ≥ 30 kg/m^2^) with pre-metabolic syndrome manifestations (such as insulin resistance, dyslipidemia, etc.) were recruited and randomly divided into a HMO supplementation group and placebo group. The HMO supplementation group received daily nutritional supplements containing specific HMOs for 24 weeks, during which the changes in body weight, body fat rate, blood glucose, blood lipid, and other indicators were monitored, and the composition of intestinal microorganisms was analyzed. The results showed that the average weight of the patients in the HMOs supplement group decreased by 3.5 kg, and the body fat rate decreased by 2.5%, which was significantly better than that of the placebo group. In terms of blood glucose indicators, fasting blood glucose and 2 h postprandial blood glucose were reduced by 0.8 mmol/L and 1.2 mmol/L, respectively, and the insulin resistance index (HOMA-IR) improved by about 20%; in the lipid profile, triglyceride levels decreased by 0.3 mmol/L, and high-density lipoprotein cholesterol (HDL-C) levels increased by 0.1 mmol/L [66].

Although HMO supports the infant gut microbiota, the gut microbiota helps to activate peripheral immune cells in the intestine. Interestingly, these immune cells may play a role in regulating the body’s response to neurogenesis (production of nerve cells). In preclinical and clinical studies, researchers have shown that HMOs can indirectly support cognitive development, promote hippocampal development and memory formation, and contribute to long-term strengthening of synapses by affecting the intestinal and immune systems. Synapses are connections between neurons. In addition, the gut microbiota is also believed to affect the production of neurotransmitters (chemicals that transmit information between brain neurons), which provides additional support for the possibility that HMO may affect the growth of protective symbiotic bacteria in the infant’s intestine during the critical period of development and, thus, may affect the growth and function of the brain [67]. The gut–brain axis is the signaling that takes place between the GI tract and the central nervous system and has been a topic of research interest in recent years. The bacterial composition of the GI tract has been linked to changes in the brain and behavior, particularly with respect to cognitive function [68]. One mechanism through which HMOs may promote neurocognitive development is by enhancing the growth of specific gut bacteria that produce SCFAs. SCFAs are able to cross the blood–brain barrier, which allows them to directly interact with, for example, microglial cells. SCFAs are proposed to regulate microglial functions that are disrupted in Alzheimer’s disease [69]. Additionally, HMOs may strengthen intestinal barrier permeability due to supporting beneficial gut microbial composition associated with their consumption. Increased intestinal barrier permeability has been associated with cognitive disorders like schizophrenia, highlighting the link between neurocognition and the GI tract [70]. Thus, as HMOs can alter the microbiome and the intestinal barrier, it can be hypothesized that HMOs might influence infant neurocognition. Although schizophrenia and Alzheimer’s are not prevalent in infants, it might very well be that HMOs are important and contribute to the prevention of these diseases in later life. A correlation has been observed between non-breastfed infants and the likelihood of developing depression in later life [71]. In infants, studies have shown beneficial effects of HMOs on cognitive development. For example, higher concentrations of 2′-FL in breastmilk were associated with better cognitive performance in infants at 24 months of age [72]. Additionally, sialic acid concentrations in the brains of breastfed newborns were substantially greater than in formula-fed infants, suggesting differences in neurodevelopment, Furthermore, total and individual fucosylated and sialylated HMOs were positively associated with cognitive, language, and motor skill domains between 18 and 24 months of age [67]. Other studies have shown that hMOs enhance the absorption of nutrients such as iron and zinc, which are critical for brain development and further support the role of hMOs in brain development. Though the number of studies in infants observing a positive link between cognition and hMOs are still limited, the proof is strengthened by effects observed in animal models [73].

## 6. Prospects for Future Research Directions

With the rapid development of science and technology, especially the vigorous rise in multi-omics technology, our understanding of the interaction between HMOs (human milk oligosaccharides) and intestinal microorganisms is deepening. Multi-omics technologies, including metagenomics, metatranscriptomics, metaproteomics, and metabolomics, provide us with a powerful tool for comprehensive and in-depth analysis of intestinal microbial community structure, function, gene expression, and metabolite changes. These technologies enable us to track the metabolic trajectory of HMOs in the intestine, interpret the interaction between HMOs and intestinal microorganisms in detail from the molecular level, and draw a more accurate regulatory network map, providing a solid theoretical basis for personalized nutrition intervention and disease prevention and control strategies.

Based on the profound impact of HMOs on intestinal microorganisms, the development of functional foods or nutritional supplements has become an important direction for future research. Using modern biotechnology, such as gene editing and synthetic biology, we can accurately design and optimize the production strains and synthesis processes of HMOs to achieve efficient and low-cost production of HMOs. Based on the successful cases in the past few decades, researchers have developed a variety of HMOs synthesis pathways based on the metabolic engineering of E.coli BL21 (DE3). In addition, the European Commission implemented regulations to approve the use of E.coli BL21 (DE3) in the production of HMOs. The development of HMO biosynthetic strains is usually carried out through the design-build-test-learn (DBTL) cycle. The first step in the production of HMOs by E.coli BL21 (DE3) is to determine its metabolic pathway. The second step is to design the metabolic pathway of precursor molecules to HMOs. Next, metabolic pathways are constructed using genetic engineering tools. Finally, the transformation level of E.coli BL21 (DE3) precursor molecules to HMOs are detected by plasmid expression or chromosome integrated expression pathway genes [74]. By constructing a model, the researchers described the most likely pathway for the synthesis of oligosaccharides that account for >95% of the HMO content in breast milk. Through the model, the candidate genes of HMO extension, branching, fucosylation, and sialylation were proposed. The model polymerization method restored two of the two previously known gene–enzyme relationships and two of the three empirically confirmed gene–enzyme relationships. These results provide the molecular basis for HMO biosynthesis to guide the progress of HMO research and application to understand and improve infant health and development [36]. In recent years, 2′-fucosyllactose (2′-FL) and lacto-N-neotetraose (LNnT) have been officially approved as food ingredients. Infant formula supplemented with these HMOs has good tolerance. However, more prospective clinical studies are needed to elucidate the importance of HMO in infant nutrition. Breastfeeding is still the best choice for infant nutrition and development. When breast milk is insufficient or unavailable, HMO-added infant formula can be considered as an alternative. The combination of preclinical and clinical cohort studies may help to determine whether individual HMO contributes to disease protection. In recent years, 2′-fucosyllactose (2′-FL) and lacto-N-neotetraose (LNnT) have been officially approved as food ingredients. Infant formula supplemented with these HMOs has good tolerance. However, more prospective clinical studies are needed to elucidate the importance of HMO in infant nutrition. Breastfeeding is still the best choice for infant nutrition and development. When breast milk is insufficient or unavailable, HMO-added infant formula can be considered as an alternative. The combination of preclinical and clinical cohort studies may help to determine whether individual HMO contributes to disease protection [5]. Combined with the research results of intestinal microbiome, according to the intestinal microecological characteristics, physiological needs, and health status of different populations, personalized nutritional formulations were tailored to develop a series of functional foods or nutritional supplements with specific efficacy. These products can not only meet the special nutritional needs of infants and young children’s growth and development but also provide accurate nutritional support and disease adjuvant treatment programs for specific groups such as patients with intestinal diseases and the elderly, so as to help them recover their health and improve their quality of life.

In the field of prevention and treatment of intestinal diseases, HMOs have great potential. Future research should focus on how to combine HMOs with existing treatment methods to carry out more in-depth clinical research and practical exploration. Through large-scale, multi-center clinical trials, the safety and efficacy of HMOs in the prevention and treatment of intestinal inflammation, infection, tumor, and other diseases are comprehensively evaluated, and rich clinical evidence is accumulated, which opens up a broad road for its clinical application. At the same time, in-depth exploration of the mechanism of HMOs regulating intestinal immunity and repairing intestinal barrier function can inject new vitality into the pathogenesis of intestinal diseases and provide new targets and ideas for innovative drug development. We have reason to believe that in the near future, HMOs are expected to become a bright pearl in the field of intestinal disease prevention and control, bring new health dawn to countless patients, and promote human health to a new peak.

## Figures and Tables

**Figure 1 nutrients-17-00519-f001:**
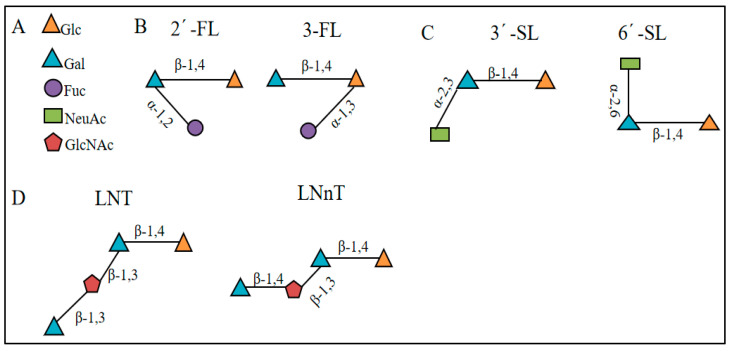
The structure of some HMOs is shown in (**A**): Five glycosyl groups constituting HMOs. (**B**): Fucosylated HMOs. (**C**): Galactosyl HMOs. (**D**): HMOs containing type I or type I chai.

**Figure 2 nutrients-17-00519-f002:**
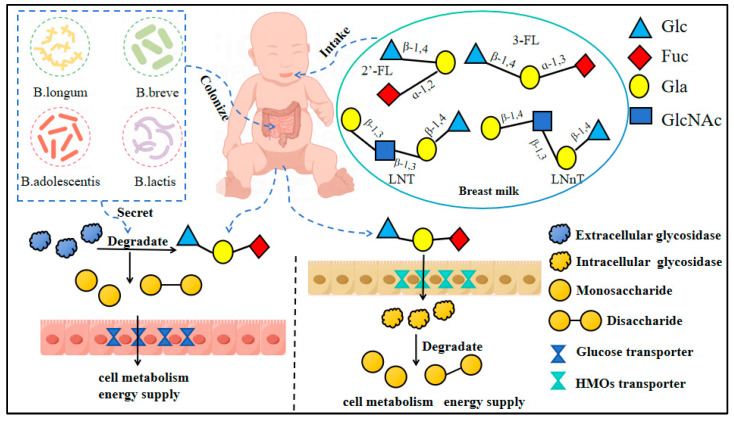
The degradation mechanism map of HMOs by Bifidobacteria.

**Table 1 nutrients-17-00519-t001:** Enzymes involved in the degradation of HMOs.

Name of Enzyme	Degradation Products of HMOs	Reference
Fucosidase	2′-FL, LNFPI, LNFPII, LNFP III, LDFT, LNDFHI, 3-FL	[32,40]
NanH2	3′-SL, 6′-SL	[31,32]
β-galactosidase	LNT, LNB, LNH, LNnt, LNTri-Ⅱ	[31,32,41]
β-n-acetylgalactosaminidase	LNTriⅡ, LNH, GlcNAC	[32,42]
GNB/LNB phosphorylase	LNB	[33]

**Table 2 nutrients-17-00519-t002:** Utilization of HMOs by some intestinal bacteria.

Types of Gut Bacteria	Using the Characteristics of HMOs	Using the Types of HMOs	Reference
Bifidobacterium breve	A variety of HMOs are not available.degradable. Can use short-chain oligosaccharides	LNB, LNT, LNnT	[44,56]
Bifidobacterium bifidum	It can degrade a variety of HMOs.but the degradation ability is medium.	LNB, LNT, LNnT, 2′-FL, 3′-FL	[44,57]
Bifidobacterium infantis	It can utilize a wide range of HMOs, preferentially consume high fucosylated structures with high degree of polymerization, prefer shorter	LNB, LNT, LNnT, 2′-FL, 3′-FL, LNFPIII	[44,57,58]
Bifidobacterium longum	A variety of HMOs are not available.degradable. Can use short-chain oligosaccharides	LNB, LNT, LDFT, 2′-FL, 3′-FL, 3′-SL, 6′-SL	[57,58,59]
Bifidobacterium adolescens	No degradation of HMOs	—	[59]
Bifidobacterium animalis	No degradation of HMOs	—	[59]
Bacteroides fragilis	Degradable HMOs of all structures	LNB, LNT, LDFT, 2′-FL, 3′-FL, 3′-SL, 6′-SL, LNFPIII, LNnt	[59,60]
Bacteroides thetaiotaomicron	Degradable HMOs of all structures	LNB, LNT, LDFT, 2′-FL, 3′-FL, 3′-SL, 6′-SL, LNFPIII, LNnt	[59,60]

**Table 3 nutrients-17-00519-t003:** Nutrient composition and content of different infant milk powder.

Name of Infant Formula	Added Nutrients	Content (%)
Aptamil German Platinum Edition	HMOs, GOS, FOS	1.22%
Abbott infant milk powder	HMOs	1.8%
Golden Crown Treasure Care Milk Powder	HMOs, lactoferrin, active protein OPN	1–1.3%
MENGNIU Future Star	HMOs, GOS, FOS α-Whey Protein and 10+ Nutrients	2–2.5%
Wyeth Revelation Blue Diamond 2	HMOs	0.72%
Love Tammy miracle blue jar	HMOs, GOS, FOS	0.32%
Flying Crane Star Flying Zhuo Rui	OPO	-
Enlightening the future	HMO, OPO	0.5%
Dutch version Meisujiaer	HMO, Novas	0.41%
BEBA Love His Beauty Baba Supreme Edition	HMO and 70% protein	0.32%

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
