# Peer review of "Research Progress on the Degradation of Human Milk Oligosaccharides (HMOs) by Bifidobacteria"

_nutrients, 2025, doi:10.3390/nu17030519_

Round 1
Reviewer 1 Report
Comments and Suggestions for Authors
The manuscript focuses on the degradation mechanism of human milk oligosaccharides (HMOs) by Bifidobacterium and its impact on infant nutrition and health. The manuscript offers a valuable contribution to infant nutrition and gut microbiota research. It effectively summarizes current knowledge from various sources and provides a comprehensive understanding of HMOs, their degradation by Bifidobacterium, and their impact on infant health. However, some sections lack specificity in identifying key research questions. The manuscript could benefit from additional examples of specific applications of HMOs. A more detailed discussion of the limitations and challenges associated with HMO research and applications would improve the scientific rigor of the manuscript.
· Abstract – Instead of making a general statement about the impact on infant health, it would be helpful to identify specific benefits. The importance of these findings for clinical practice should also be emphasized.
· 1. Introduction – Provide a more detailed explanation of HMOs and their importance in infant gut health. While the purpose of the study is mentioned, the scope of the review could be more specific. It also emphasizes the importance of understanding HMO degradation by Bifidobacterium. More recent and relevant studies should be cited in the introduction to present the current state of research in this area.
· 2. The Composition and Properties of HMOs – The information on monosaccharides and HMO properties could be presented more broadly. The definition of HMOs can be omitted. While factors affecting the composition and content of HMOs are mentioned in the text, this could be explained in more detail. Concrete examples would be beneficial. The text lacks references to recent research on HMOs.
· 3. The Degradation Mechanism of HMOs by Bifidobacterium – The text should specifically refer to Figure 2 and Table 1, explain their meaning, and highlight the key information. It would be helpful to study the effects of HMO depletion on infant health in more detail. The section lacks references to recent research on the mechanisms of HMO degradation.
· 4. Effects of HMOs on Gut Microbiota – Rather than focusing on beneficial and harmful bacteria separately, subsections 4.1 and 4.2 could be combined to provide a more coherent account of the effects of HMOs on different bacterial populations in the gut. Expanding the description of the mechanisms by which HMOs affect the gut microbiota is worth expanding. The text should clearly reference Table 2, which provides detailed information on utilizing HMOs by different types of gut bacteria. To strengthen the message and increase the text's credibility, it is worth adding examples of clinical studies confirming the positive effects of HMOs on the intestinal microbiota in infants. The bibliography should be updated with the latest scientific publications to reflect the current state of knowledge in the field.
· 5. Effect of HMOs Formula Milk Powder on Intestinal Flora of Infants – To avoid redundancies, the descriptions of the two clinical trials could be summarized in a single paragraph. It is worth noting that different infant formulas may contain different types and amounts of HMOs. It would be beneficial to include specific data from the studies or at least clarify which aspects of brain development could potentially be supported by HMOs. In the text, specific references should be made to Table 2, which shows various infant formulas' nutritional composition and content. It is recommended that the bibliography be updated with the latest scientific publications on the effects of HMO-fortified infant formula on infant gut microbiota. Consider adding information about the possible limitations of the studies cited.
· 6. Prospects for Future Research Directions – The chapter mentions oligosaccharide technologies but does not elaborate on their specific application in HMO research. While the chapter describes the potential uses of HMOs in functional foods and dietary supplements, concrete examples are missing. Consider a more detailed discussion of the potential of HMOs in preventing and treating intestinal diseases. The chapter focuses on the potential of HMOs, but it would also be useful to mention possible challenges and limitations related to HMO research and applications. I propose a more precise definition of future research directions. As in the previous sections, it is worth updating the bibliography with the latest scientific publications to reflect current knowledge about the future directions of HMO research.
Author Response
Comments:Abstract – Instead of making a general statement about the impact on infant health, it would be helpful to identify specific benefits. The importance of these findings for clinical practice should also be emphasized. |
Response: Thank you for pointing this out. We agree with this comment. Therefore, we have made the following modifications. The effects of HMOs formula milk powder on the intestinal microbiota of infants were discussed, including simulating breast milk composition, regulating intestinal flora and immune function, infection prevention and brain development. Finally, the research results are summarized and future research directions are proposed to provide directions for research in the field of infant nutrition. It can be found in lines 13-18 of the revised manuscript. |
Comments 1: Introduction – Provide a more detailed explanation of HMOs and their importance in infant gut health. While the purpose of the study is mentioned, the scope of the review could be more specific. It also emphasizes the importance of understanding HMO degradation by Bifidobacterium. More recent and relevant studies should be cited in the introduction to present the current state of research in this area. |
Response 1: Agree. We have done modified partial contents to emphasize this point. HMO plays an irreplaceable role in infant intestinal health. In the construction of intestinal flora, after the birth of the baby, the intestine is a relatively sterile environment, which needs to quickly establish a beneficial microbial community. HMO can accurately provide nutrients for the growth of beneficial bacteria such as bifidobacteria and stimulate their mass reproduction[6]. For example, bifidobacteria longum and bifidobacteria shortum can use HMO for growth and metabolism, thus forming dominant flora in the intestine. These beneficial bacteria inhibit the growth and colonization of harmful bacteria such as Escherichia coli and Salmonella by competing for nutrients and producing antibacterial substances, maintain the balance of intestinal microecology, and effectively reduce the risk of intestinal infection in infants[7]. By degrading HMO, Bifidobacterium can convert it into metabolites such as short-chain fatty acids ( such as lactic acid and acetic acid ). These metabolites not only provide energy for intestinal epithelial cells, but also reduce intestinal pH and inhibit the growth of harmful bacteria, thereby maintaining the balance of intestinal flora. At the same time, it is of great significance to the development of infant immune system. Studies have shown that bifidobacteria produce metabolites such as aromatic lactic acid during the degradation of HMO.These substances can regulate the function of immune cells, enhance the immune response of infants, and reduce the risk of infection[9]. Bifidobacterium can also increase the production of secretory immunoglobulin A ( sIgA ) by promoting the colonization of intestinal beneficial bacteria, and further enhance the immune barrier function of intestinal mucosa[10]. In the process of breastfeeding, the efficient use of HMO by Bifidobacterium makes it occupy a competitive advantage in the infant 's intestine, which helps infants better adapt to breast milk nutrition[11]. This symbiotic relationship not only optimizes the intestinal environment, but also plays a key role in the early intestinal barrier function and immune system development of infants[12]. It can be found in lines 38-47 57-72 of the revised manuscript.
|
Comments 2: The Composition and Properties of HMOs – The information on monosaccharides and HMO properties could be presented more broadly. The definition of HMOs can be omitted. While factors affecting the composition and content of HMOs are mentioned in the text, this could be explained in more detail. Concrete examples would be beneficial. The text lacks references to recent research on HMOs. |
Response 2: Agree. We have done modified partial contents to emphasize this point. HMOs are naturally occurring oligosaccharides with a degree of polymerization not higher than 3 in breast milk, and their monosaccharide composition is rich and diverse[19]. According to the different structural types, HMOs are mainly divided into neutral fucosylated HMOs, such as 2' -fucosyllactose ( 2'-FL ) and 3-fucosyllactose ( 3-FL ), neutral non-fucosylated HMOs, such as lacto-N-tetraose ( LNT ) and lacto-N-neotetraose ( LNnT ), and acidic sialylated HMOs. Such as 3 ' -sialyllactose ( 3' -SL ) and 6' -sialyllactose ( 6' -SL )[20]. After ingestion of HMOs, about 0.05 % of HMOs enter the blood, and most of the remaining HMOs enter the large intestine. Observational studies have shown that HMOs play an important role in improving intestinal flora microecology, maintaining intestinal barrier, regulating immunity, resisting pathogen infection and promoting neural development[21]. It is mainly modified by five monomers glucose, galactose, N-acetylglucosamine, fucose and N-acetylneuraminic acid. Among them, D-glucose ( Glc ) is a common monosaccharide that plays an important structural support role in HMOs. D-galactose ( Gal ) is similar to glucose in structure and participates in the construction of complex structures of HMOs. N-acetylglucosamine ( GlcNAc ) brings unique chemical properties to HMOs[22]. L-fucose ( Fuc ) endows HMOs with specific biological functions, while N-acetylneuraminic acid ( Neu5Ac ) in sialic acid ( Sia ) is the most important form of sialic acid and plays a key role in the function of HMOs[23]. These monosaccharides are linked to lactose through different glycosidic bonds to form different structures of HMOs. Each HMOs contains a lactose at the reducing end, and on this basis, the galactose β-1,3-N-acetylglucosamine is connected by β-1,3 or β-1,6 bonds or the N-acetylgalactosamine is connected to extend the sugar chain to form core structures such as lacto-N-tetrose ( LNT ) and lacto-N-neotetraose ( LNnT )[24]. Breast milk oligosaccharides are complex mixtures of many oligosaccharides, and more than 200 HMO structures have been isolated and identified[25].The structure of some HMOs is shown in Fig.1. It can be found in lines 100-108 of the revised manuscript. |
Comments 3: The Degradation Mechanism of HMOs by Bifidobacterium – The text should specifically refer to Figure 2 and Table 1, explain their meaning, and highlight the key information. It would be helpful to study the effects of HMO depletion on infant health in more detail. The section lacks references to recent research on the mechanisms of HMO degradation.
Response 3: Agree. We have done modified partial contents to emphasize this point.
Bifidobacteria play an important role in the metabolism of HMOs in infant gut microbiota. It has been found that these bacteria have a variety of genes that encode glycosidases and transporters related to HMOs metabolism[28]. B. longum subsp strains possess a 43-kb gene cluster specific for HMO degradation[29]. This gene cluster encodes all GH enzymes required to efficiently cleave HMOs including 1,2-α-fucosidases belonging to the GH95 family, 1,3/4-α-fucosidases from GH29, 2,3/6 sialidases belonging to the GH33 family, β-N-acetylhexosaminidase enzymes from the GH20 family, β-galactosidases from the GH2 family and LNT β-galactosidases from the GH42 family[29-30]. Interestingly, these enzymes appear to be intracellular due to the absence of an N-terminal signal sequence in their primary structure[29]. This gene cluster also includes several genes which encode sugar transporters involved in importing HMOs intact such as ATP-binding cassette (ABC) transporters, GNB/LNB pathway transporters as well as solute binding proteins (SBPs)[31]. ABC transporters recognize and bind to HMOs through their extracellular solute-binding proteins ( SBPs ). These SBPs have high substrate specificity and can bind to the specific structure of HMOs and transmit them to the membrane protein complex. When SBPs bind to HMOs, the membrane protein portion of ABC transporters undergoes conformational changes, binds to ATP and hydrolyzes ATP to release energy, thereby transporting HMOs into cells. This transport process not only depends on the high specificity of ABC transporters, but also is closely related to the structural complexity of HMOs[32]. SBPs interact with specific glycan structures in HMOs through their binding sites, thereby realizing the recognition and binding of HMOs. This binding depends on the highly specific structure of SBPs, enabling them to form stable interactions with specific glycans such as fucose, sialic acid or lactose in HMOs. During the binding process, the structure of SBPs will undergo conformational changes, which will help to further enhance the binding stability with HMOs. This high affinity binding allows SBPs to introduce intact HMOs molecules into cells through ABC transporters[33].
It can be found in lines 160-187 of the revised manuscript.
Comments 4: Effects of HMOs on Gut Microbiota – Rather than focusing on beneficial and harmful bacteria separately, subsections 4.1 and 4.2 could be combined to provide a more coherent account of the effects of HMOs on different bacterial populations in the gut. Expanding the description of the mechanisms by which HMOs affect the gut microbiota is worth expanding. The text should clearly reference Table 2, which provides detailed information on utilizing HMOs by different types of gut bacteria. To strengthen the message and increase the text's credibility, it is worth adding examples of clinical studies confirming the positive effects of HMOs on the intestinal microbiota in infants. The bibliography should be updated with the latest scientific publications to reflect the current state of knowledge in the field.
Response 4: Agree. We have done modified partial contents to emphasize this point.
Studies have found that Bifidobacterium infantis in newborns can inhibit the growth of harmful microorganisms in the intestine and help infants digest breast milk oligosaccharides ( HMO ) in breast milk. When HMO enters the infant intestine, Bifidobacterium infantis can convert it into short-chain fatty acids, thereby effectively reducing the pH value of infant feces and creating an acidic environment that is not conducive to the growth of harmful bacteria. As the baby grows, this probiotic will gradually decrease, but if it does not appear in the early stage, other bacteria, especially pathogens, may fill its growth space, further increasing the probability of infants suffering from related diseases[48].Researchers analyzed breast milk oligosaccharides in 33 children with necrotizing enterocolitis ( NEC ) and 37 controls, and performed longitudinal metagenomic sequencing of feces from 48 infants, including 14 children with NEC. The results showed that the concentrations of HMO and DSLNT in breast milk of mothers with NEC were significantly lower than those in the control group. The sensitivity and specificity of MOM threshold of 241 nmol / ml for NEC was 0.9. The metagenomic sequencing before the onset of NEC showed that the relative abundance of Bifidobacterium longum decreased and the relative abundance of Enterobacter cloacae increased in children with NEC. Low MOM DSLNT affects the longitudinal development of the microbiome, which is associated with a decrease in the type of gut microbiota in preterm infants transitioning to bifidobacteria-dominated preterm infants, usually observed in older infants. By combining pre-disease HMO and metagenomic data with random forest analysis, the accuracy of distinguishing infant health or NEC reached 87.5 %[49].The researchers compared the composition of intestinal flora between cesarean section and natural delivery infants, and found that the abundance of beneficial bacteria in the intestinal flora of cesarean section infants was lower, while the relative proportion of harmful bacteria was higher. Further studies have found that in breast-fed cesarean section infants, if breast milk contains abundant HMO such as 2'-FL, the structure and function of the infant 's intestinal flora will be closer to that of natural birth infants. 2'-FL can promote the growth and colonization of beneficial bacteria such as Bifidobacterium.Bifidobacteria produce short-chain fatty acids by fermenting HMO, reduce intestinal pH, inhibit the growth of harmful bacteria, and regulate the balance of intestinal flora[50].In the study of children with milk protein allergy ( CMPA ), deep hydrolyzed formula milk powder supplemented with HMO was used for feeding. Studies have found that HMO can regulate intestinal flora, promote the enrichment of beneficial bacteria such as Bifidobacterium, and reduce the number of potentially harmful bacteria such as Escherichia coli. For children with CMPA who began to be fed with a deeply hydrolyzed formula ( eHF ) containing HMO before 3 months of age, this regulatory effect is more obvious, which helps to reverse the state of intestinal flora imbalance[51].In related studies, the intestinal environment was simulated by in vitro experiments, and common intestinal pathogens, such as Escherichia coli and Salmonella, were co-cultured with culture medium containing HMO. The results showed that HMO could bind to the adhesion protein on the surface of pathogens and block the adhesion of pathogens to intestinal epithelial cells, thus effectively inhibiting the colonization and reproduction of pathogens in the intestine[52].Researchers, by studying the isolated colonic smooth muscle tissue, applied different concentrations of fucose-based HMO, and used a tension sensor to accurately measure the changes in the contractility of the smooth muscle. The results showed that with the increase of fucosylated HMO concentration, the contractility of colonic smooth muscle gradually decreased, and this effect showed a certain dose-dependent manner[53].
It can be found in lines252-300 of the revised manuscript.
Comments 5: Effect of HMOs Formula Milk Powder on Intestinal Flora of Infants – To avoid redundancies, the descriptions of the two clinical trials could be summarized in a single paragraph. It is worth noting that different infant formulas may contain different types and amounts of HMOs. It would be beneficial to include specific data from the studies or at least clarify which aspects of brain development could potentially be supported by HMOs. In the text, specific references should be made to Table 2, which shows various infant formulas' nutritional composition and content. It is recommended that the bibliography be updated with the latest scientific publications on the effects of HMO-fortified infant formula on infant gut microbiota. Consider adding information about the possible limitations of the studies cited.
Response 5: Agree. We have done modified partial contents to emphasize this point.
Although HMO supports the infant gut microbiota, the gut microbiota helps to activate peripheral immune cells in the intestine. Interestingly, these immune cells may play a role in regulating the body 's response to neurogenesis ( production of nerve cells ). In preclinical and clinical studies, researchers have shown that HMO can indirectly support cognitive development, promote hippocampal development and memory formation, and contribute to long-term strengthening of synapses by affecting the intestinal and immune systems. Synapses are connections between neurons. In addition, the gut microbiota is also believed to affect the production of neurotransmitters ( chemicals that transmit information between brain neurons ), which provides additional support for the possibility that HMO may affect the growth of protective symbiotic bacteria in the infant 's intestine during the critical period of development, and may thus affect the growth and function of the brain[73].The gut–brain-axis is the signaling that takes place between the GI tract and the central nervous system and has been a topic of research interest in recent years. The bacterial composition of the GI tract has been linked to changes in the brain and behavior, particularly with respect to cognitive function[74].One mechanism through which HMOs may promote neurocognitive development is by enhancing the growth of specific gut bacteria that produce SCFAs. SCFAs are able to cross the blood–brain barrier, which allows them to directly interact with, for example, microglial cells. SCFAs are proposed to regulate microglial functions that are disrupted in Alzheimer’s disease[75]. Additionally, HMOs may strengthen intestinal barrier permeability due to supporting beneficial gut microbial composition associated with their consumption. Increased intestinal barrier permeability has been associated with cognitive disorders like schizophrenia highlighting the link between neurocognition and the GI tract[76]. Thus, as hMOs can alter the microbiome and the intestinal barrier, it can be hypothesized that hMOs might influence infant neurocognition. Although schizophrenia and Alzheimer’s are not prevalent in infants, it might very well be that hMOs are important and contribute to the prevention of these diseases in later life. A correlation has been observed between non-breastfed infants and the likelihood of developing depression in later life[77].In infants, studies have shown beneficial effects of hMOs on cognitive development. For example, higher concentrations of 2′-FL in breastmilk were associated with better cognitive performance in infants at 24 months of age[78]. Additionally, sialic acid concentrations in the brains of breastfed newborns were substantially greater than in formula-fed infants, suggesting differences in neurodevelopment, Furthermore, total and individual fucosylated and sialylated hMOs were positively associated with cognitive, language, and motor skill domains between 18 and 24 months of age[73]. Other studies have shown that hMOs enhance the absorption of nutrients such as iron and zinc, which are critical for brain development and further support the role of hMOs in brain development. Though the number of studies in infants observing a positive link between cognition and hMOs are still limited, the proof is strengthened by effects observed in animal models[79].
It can be found in lines 389-429 of the revised manuscript.
Comments 6: Prospects for Future Research Directions – The chapter mentions oligosaccharide technologies but does not elaborate on their specific application in HMO research. While the chapter describes the potential uses of HMOs in functional foods and dietary supplements, concrete examples are missing. Consider a more detailed discussion of the potential of HMOs in preventing and treating intestinal diseases. The chapter focuses on the potential of HMOs, but it would also be useful to mention possible challenges and limitations related to HMO research and applications. I propose a more precise definition of future research directions. As in the previous sections, it is worth updating the bibliography with the latest scientific publications to reflect current knowledge about the future directions of HMO research.
Response 6: Agree. We have done modified partial contents to emphasize this point.
Based on the successful cases in the past few decades, researchers have developed a variety of HMOs synthesis pathways based on the metabolic engineering of E.coli BL21 ( DE3 ). In addition, the European Commission implemented regulations to approve the use of E.coli BL21 ( DE3 ) in the production of HMOs. The development of HMO biosynthetic strains is usually carried out through the design-build-test-learn ( DBTL ) cycle. The first step in the production of HMOs by E.coli BL21 ( DE3 ) is to determine its metabolic pathway. The second step is to design the metabolic pathway of precursor molecules into hmo. Next, metabolic pathways were constructed using genetic engineering tools. Finally, the transformation level of E.coli BL21 ( DE3 ) precursor molecules to HMOs was detected by plasmid expression or chromosome integrated expression pathway genes[90]. By constructing a model, the researchers described the most likely pathway for the synthesis of oligosaccharides that account for > 95 % of the HMO content in breast milk. Through the model, the candidate genes of HMO extension, branching, fucosylation and sialylation were proposed. The model polymerization method restored two of the two previously known gene-enzyme relationships and two of the three empirically confirmed gene-enzyme relationships. These results provide the molecular basis for HMO biosynthesis to guide the progress of HMO research and application to understand and improve infant health and development[91].In recent years, 2 ' -fucosyllactose ( 2 ' -FL ) and lacto-N-neotetraose ( LNnT ) have been officially approved as food ingredients. Infant formula supplemented with these HMOs has good tolerance. However, more prospective clinical studies are needed to elucidate the importance of HMO in infant nutrition. Breastfeeding is still the best choice for infant nutrition and development. When breast milk is insufficient or unavailable, HMO-added infant formula can be considered as an alternative. The combination of preclinical and clinical cohort studies may help to determine whether individual HMO contributes to disease protection. In recent years, 2'-fucosyllactose (2'-FL ) and lacto-N-neotetraose ( LNnT ) have been officially approved as food ingredients. Infant formula supplemented with these HMOs has good tolerance. However, more prospective clinical studies are needed to elucidate the importance of HMO in infant nutrition. Breastfeeding is still the best choice for infant nutrition and development. When breast milk is insufficient or unavailable, HMO-added infant formula can be considered as an alternative. The combination of preclinical and clinical cohort studies may help to determine whether individual HMO contributes to disease protection[92].
It can be found in lines 447-480 of the revised manuscript.
Reviewer 2 Report
Comments and Suggestions for Authors
Summary statement
Running title: Research Progress on the Degradation of HMOs 2 by Bifidobacteria
The manuscript aims to elucidate the degradation mechanism of Bifidobacterium on breast milk oligosaccharides (HMOs ) and its application in infant nutrition and health.
The review is relevant and also has an interesting and important approach, but we suggest that the authors review a few points:
1. Introduction
In line 21 you don't need to repeat “breast milk, breast milk oligosaccharides (HMOs)”
In line 23 we recommend that you standardize the objective. The objective must be the same as it is in the abstract.
In line 44 put the reference right after the phrase "The other is extracellular digestion”.
In lines 48, 49 and 50 please check if the information is correct. “In addition, the ability of Bifidobacterium to utilize HMOs is also different. B.infantis has a strong ability to degrade HMOs. Although B.infantishas less ability to use HMOs than B.infantis, it can still degrade HMOs into monosaccharides”.
2. The Composition and Properties of HMOs
2.2. Performance of HMOs
In lines 89 and 90 could you explain better when you talk about non-secretory mothers. In this case, are these women unable to produce breast milk?
In line 104 put the reference right after the phrase “The content of different types of HMOs in breast milk is different”.
In lines 112, 113, and 144 I suggest that you remove the word “conclusion”. We think it is too hasty to include it in this part of the text. The conclusion should go at the end of the article. You can state that the reviewed scientific literature shows that the main components of HMOs play an important role in child growth and development.
3. The Degradation Mechanism of HMOs by Bifidobacterium
Please cite the Figure 2 and Table 1 in the text.
4. Effects of HMOs on Gut Microbiota 173
4.1. Promote the Growth of Beneficial Bacteria
In line 186 please put the reference right after the paragraph “Lactic acid can not only reduce the local pH value of the intestine, inhibit the growth of harmful bacteria such as Escherichia coli and Salmonella, but also promote the absorption of calcium, iron, phosphorus and other minerals in the intestine, and provide support for the skeletal development and overall health of infants and young children”
5. Effect of HMOs Formula Milk Powder on Intestinal Flora of Infants
Please cite the Table 2 in the text.
In Table 2 we recommend that more infant formulas be cited, so that it does not appear that the authors have a conflict of interest.
Author Response
|
|
|
3. Point-by-point response to Comments and Suggestions for Authors |
||
Comments 1: In line 21 you don't need to repeat “breast milk, breast milk oligosaccharides (HMOs)” In line 23 we recommend that you standardize the objective. The objective must be the same as it is in the abstract. In line 44 put the reference right after the phrase "The other is extracellular digestion”. In lines 48, 49 and 50 please check if the information is correct. “In addition, the ability of Bifidobacterium to utilize HMOs is also different. B.infantis has a strong ability to degrade HMOs. Although B.infantishas less ability to use HMOs than B.infantis, it can still degrade HMOs into monosaccharides”. |
||
Response 1: Thank you for pointing this out. We agree with this comment. Therefore, we have made the following modifications. In line 21 you don't need to repeat “breast milk, breast milk oligosaccharides (HMOs)” HMOs are a variety of complex carbohydrates that play a key role in the growth and development of infants. They are the third most abundant solid component in breast milk, with more than 200 different types, providing a variety of health benefits, such as promoting digestive health, regulating immune response, and promoting cognitive development[1]. Bifidobacterium is one of the most abundant microbial genera in the intestine of breastfed infants and plays an important role in the metabolism of HMOs.Breast milk is the gold standard of infant nutrition, which is rich in HMOs[2]. It can be found in lines 22-26 of the revised manuscript. In line 23 we recommend that you standardize the objective. The objective must be the same as it is in the abstract. In the Abstract, I wrote : The purpose of this study was to investigate the degradation mechanism of Bifidobacterium on breast milk oligosaccharides ( HMOs ) and its application in infant nutrition.The 23 line is consistent with the abstract after modification. It can be found in the revised manuscript. In line 44 put the reference right after the phrase "The other is extracellular digestion”. One is intracellular digestion, and Bifidobacterium breve, Bifidobacterium infantis, and LnbX-negative Bifidobacterium longum directly transfer complete HMOs into cells through specific transporters for degradation. The other is extracellular digestion. Bifidobacterium bifidum and LnbX-positive Bifidobacterium longum secrete extracellular glycosidases to degrade extracellular HMOs, and then the released monosaccharides or disaccharides are introduced into the cells for further degradation[9]. It has been modified in the original text. In lines 48, 49 and 50 please check if the information is correct. “In addition, the ability of Bifidobacterium to utilize HMOs is also different. B.infantis has a strong ability to degrade HMOs. Although B.infantishas less ability to use HMOs than B.infantis, it can still degrade HMOs into monosaccharides”. In addition, the ability of different Bifidobacterium species to utilize HMOs varies significantly. Bifidobacterium longum subsp. infantis (B. infantis) is renowned for its exceptional capacity to degrade HMOs, making it one of the most proficient HMO utilizers among gut bacteria. This subspecies possesses a diverse array of glycoside hydrolase enzymes that enable it to break down a wide range of HMO structures, including fucosylated and sialylated forms. While other Bifidobacterium species, such as B. bifidum, also exhibit the ability to utilize HMOs, their capacity is generally lower compared to B. infantis. However, they can still degrade HMOs into monosaccharides, contributing to the overall metabolic processes in the infant gut。 It can be found the revised manuscript. |
||
Comments 2: In lines 89 and 90 could you explain better when you talk about non-secretory mothers. In this case, are these women unable to produce breast milk? In line 104 put the reference right after the phrase“The content of different types of HMOs in breast milk is different”. In lines 112, 113, and 144 I suggest that you remove the word “conclusion”. We think it is too hasty to include it in this part of the text. The conclusion should go at the end of the article. You can state that the reviewed scientific literature shows that the main components of HMOs play an important role in child growth and development. |
||
Response 2: Agree. We have, accordingly, done modified and explain to emphasize this point. In lines 89 and 90 could you explain better when you talk about non-secretory mothers. In this case, are these women unable to produce breast milk? Explain:The composition and content of HMOs are affected by maternal genetic factors ( FUT2 gene and Se gene ). The milk of secretory ( Secretor ) and non-secretor ( Non-secretor ) secretory mothers contains higher levels of α-1,2-fucosylated HMOs, such as 2 ' -fucosylated lactose ( 2 ' -FL ), lactose-N-fucosylated I ( LNFP-I ), etc. The milk of non-secretory mothers lacks or only contains trace amounts of α-1,2-fucosylated HMOs, but the content of α-1,3 / 4-fucosylated HMOs ( such as 3 ' -fucosylated lactose, 3-FL ) is higher. The detailed explanation for the two is not the ability to secrete breast milk, but the difference in HMO in secreted breast milk, so it will be divided into secretory and non-secretory according to the composition and content of HMO in breast milk.
In line 104 put the reference right after the phrase“The content of different types of HMOs in breast milk is different”. We have done modified:The content of different types of HMOs in breast milk is different[26], and its content is also affected by the mother 's secretion type, gene, body mass index, gestational age, dietary habits, geographical environment and other factors.
In lines 112, 113, and 144 I suggest that you remove the word “conclusion”. We think it is too hasty to include it in this part of the text. The conclusion should go at the end of the article. You can state that the reviewed scientific literature shows that the main components of HMOs play an important role in child growth and development. We have done modified:The reviewed scientific literature shows that the main components of HMOs play an important role in child growth and development. The structure and content of the main components of HMOs are complex and diverse. Understanding the composition and content of HMOs is helpful to further study its application in infant nutrition and health. It can be found in the revised manuscript. Comments 3: Please cite the Figure 2 and Table 1 in the text. Response 3:We have done modified:As shown in Figure 2 Bifidobacterium has developed two mechanisms for the use of HMOs : one is digestion in cells through transporters ; the other is dependent on extracellular glycosidase digestion in the extracellular It can be found in the revised manuscript Table 1 lists the different Enzymes involved in the degradation of HMOs. The GNB / LNB transporter is responsible for the transport of LNB released extracellularly by lactate-N-bioenzymes and GNB released from mucin O-glycans by α-N-acetylgalactosaminidase. It can be found in the revised manuscript. Comments 4: 4.1. Promote the Growth of Beneficial Bacteria In line 186 please put the reference right after the paragraph “Lactic acid can not only reduce the local pH value of the intestine, inhibit the growth of harmful bacteria such as Escherichia coli and Salmonella, but also promote the absorption of calcium, iron, phosphorus and other minerals in the intestine, and provide support for the skeletal development and overall health of infants and young children” Response4:We have done modified:Lactic acid can not only reduce the local pH value of the intestine, inhibit the growth of harmful bacteria such as Escherichia coli and Salmonella, but also promote the absorption of calcium, iron, phosphorus and other minerals in the intestine, and provide support for the skeletal development and overall health of infants and young children.
Comments 5: |
Effect of HMOs Formula Milk Powder on Intestinal Flora of Infants
Please cite the Table3 in the text.
In Table 3 we recommend that more infant formulas be cited, so that it does not appear that the authors have a conflict of interest.
Response5:
We have done modified:, the content of HMO in different formula milk powder. A multicenter, double-blind, randomized controlled trial included hundreds of healthy full-term infants, who were randomly divided into two groups and fed with infant formula with HMOs and ordinary formula without HMOs, respectively.
Table 3: Nutrient composition and content of different infant milk powder.
Name of infant formula |
Added nutrients |
Content(%) |
|
Aptamil German Platinum Edition |
HMOs、GOS、FOS |
1.22% |
|
Abbott infant milk powder |
HMOs |
1.8% |
|
Golden Crown Treasure Care Milk Powder |
HMOs、lactoferrin、 active protein OPN、 |
1%-1.3% |
|
MENGNIU Future Star |
HMOs、GOS、FOS α-Whey Protein and 10+ Nutrients |
2%-2.5% |
|
Wyeth Revelation Blue Diamond 2 |
HMOs |
0.72% |
|
Love Tammy miracle blue jar |
HMOs、GOS、FOS |
0.32% |
|
Flying Crane Star Flying Zhuo Rui |
OPO |
- |
|
Enlightening the future |
HMO、OPO |
0.5% |
|
Dutch version Meisujiaer |
HMO、Novas |
0.41% |
|
BEBA Love His Beauty Baba Supreme Edition |
HMO and 70% protein |
0.32% |
Round 2
Reviewer 1 Report
Comments and Suggestions for Authors
The author responded to all the queries. I do not have any questions. Therefore, this manuscript may be considered for publication in this journal.